# Green Extraction Process of Food Grade C-phycocyanin: Biological Effects and Metabolic Study in Mice

Camilly Fratelli [1,2] , Monize Bürck [1,2] , Artur Francisco Silva-Neto [2], Lila Missae Oyama [2], Veridiana Vera De Rosso [2,3,*] and Anna Rafaela Cavalcante Braga [1,2,4,*]

1   Department of Biosciences, Universidade Federal de São Paulo (UNIFESP), Silva Jardim Street, 136, Vila Mathias, Santos 11015-020, Brazil
2   Department of Physiology, Universidade Federal de São Paulo (UNIFESP), Botucatu Street, 740, Vila Clementino, São Paulo 04023-062, Brazil
3   Nutrition and Food Service Research Center, Universidade Federal de São Paulo (UNIFESP), Rua Silva Jardim 136 CEP Santos, São Paulo 11015-020, Brazil
4   Department of Chemical Engineering, Universidade Federal de São Paulo (UNIFESP), Campus Diadema, Diadema, São Paulo 09972-270, Brazil
*   Correspondence: veridiana.rosso@unifesp.br (V.V.D.R.); anna.braga@unifesp.br (A.R.C.B.)

**Abstract:** This study aimed to evaluate different parameters in the green process of organic *Spirulina* biomass (SB) C-phycocyanin (C-PC) extraction to understand the impact on weight and oral glucose tolerance of C-PC extract in Swiss mice fed with a high-fat diet (HFD). The proximate composition and antioxidant activity were analyzed in *Spirulina* by-products: SB, C-PC, and Remaining biomass (RB). The protein content on a dry basis was 52.05% in SB and 61.16% in RB and 118.97 μg/g in C-PC. The antioxidant activity was equal for SB and C-PC but higher than RB. However, RB can be considered a promising ingredient, promoting the sustainable use of the whole SB. Swiss mice were distributed in five groups: control diet (CD), HFD, HFD plus *Spirulina* biomass (HFDS), HFD plus C-PC (HFDC), and HFD plus remaining biomass (HFDR). HFDS increased the delta weight of the animals and showed glucose intolerance compared to the CD and HFDC groups. The results demonstrated that the supplementation of 500 mg/kg of body weight of SB in the HFDS group did not show antiobesogenic potential with an HFD, but it is essential to conduct further studies to bring other interesting responses regarding C-PC biological in vivo effects.

**Keywords:** green chemistry; antioxidant activity; oral glucose tolerance test—OGTT



## 1. Introduction

The downstream process of intracellular bioproducts from *Arthrospira platensis*, also known as *Spirulina*, such as the phycobiliprotein C-PC, comprises several steps, including separation of the cells from the cultivation media, cell rupture, extraction method, purification steps, and refining of the product [1]. Extraction using inorganic solvents has been discussed in several studies as an efficient and alternative method for C-PC recovery. However, recently, efforts have been made to develop more sustainable processes to extract bioactive compounds, including low energy consumption and avoiding the employment of toxic chemicals [2], characteristics capable of triggering both health and environmental problems [3]. As a result, green chemistry procedures are essential in the lab and industry. A recent study intended to enhance extraction and purification of C-PC with lower energy consumption [4] used a shaker, centrifugation, and ultrasonication employing potassium phosphate buffer and obtained less purity ratio than another study [5] that used distilled water as a solvent and kept the *Spirulina* biomass without agitation (0.34 versus 0.60 purity), showing that simple methods are even greener and suitable for significant scaling up. Subsequent purification steps applying ammonium sulfate ($(NH_4)_2 SO_4$) and dialysis provide greater purity (1.6) [1].

C-PC has been widely studied due to its potential applications, which depend on its purity grade. According to Rito-Palomares et al. [6] and Lauceri et al. [7], to affirm that C-PC extract presents food grade purity (EP), this parameter must be equal to or superior to 0.7; this way, C-PC can be used as a food and cosmetics natural blue pigment. Additionally, C-PC presenting EP $\geq$ 1.5 are suitable for cosmetics; EP $\geq$ 3.9 denotes reactive grade, and EP $\geq$ 4.0 shows analytical, allowing its use as a pharmaceutical.

The applications of C-PC have been of great interest due to its proven biological effects. From the literature, we can highlight the antioxidant [1] anti-obesity [8], hepatoprotective [9], anti-inflammatory [10], neuroprotective [11], antidiabetic [12], and anticancer effects [13]. These pathological pathways are frequently related to damages in DNA, lipids, proteins, and carbohydrates caused by Reactive Oxygen Species (ROS) [14,15] originating from physiological and metabolic processes. The antioxidant activity is the most highlighted biological effect of C-PC extracted from *Spirulina* shown in the literature. The tetrapyrrole conformation [16] and the apoprotein portion [17] of the C-PC molecule are responsible for interacting with free radicals and ROS. More recently, Agrawal et al. [18] also hypothesized that the presence of amino acids along with the chromophore part of the C-PC might function as a synergetic oxidant because of their ability to chelate traces of pro-oxidative metal and regenerate the intracellular antioxidant system.

Bermejo et al. [19] verified *Spirulina*'s antioxidant activity in vitro by finding inhibition of hydroxyl radical-mediated deoxyribose degradation and peroxyl radical production. Additionally, Abdelkhalek et al. [20] hypothesized that *Spirulina* exerts a protective effect by indirectly leading to the enhancement of endogenous enzymatic antioxidants and/or directly scavenging free radicals inhibiting lipid peroxidation in vivo. Thus, C-PC prevents oxidative stress through interaction with several ROS (i.e., hydroxyl radical ($\bullet$OH)) and lipid peroxidation and enhances the endogenous enzymatic antioxidant profile in vivo [8,18]. Thus, C-PC food grade and *Spirulina* biomass consequent support health maintenance. Therefore, antioxidant activity as a biological effect is under exploration more than weight loss or obesity handling [15].

Zhao et al. [8] compared anti-obesity effects in mice fed with a high-fat diet (45% of the calories from lipids), plus 2 g/kg per diet of *Spirulina* platensis (WSP), *Spirulina* platensis protein (SPP), or *Spirulina* platensis protein hydrolysate (SPPH). The results demonstrated that while SPP was the best for lowering glucose (39.6%), SPPH was the best for weight reduction (39.8%) and total cholesterol reduction (20.8%). The authors highlighted that the difference between the effects of WSP, SPP, and SPPH may be explained by the compositional differences in protein fraction. The study also pointed out a modulatory effect of SPPH on the liver–brain axis. It is worth mentioning that SPPH was obtained through hydrolysis with pepsin at pH 2.0 for 10 h, followed by enzyme inactivation at 90 °C for 15 min, which most likely impacted the C-PC content [14].

C-PC is not associated with acute (2 g/kg body weight (BW) ($w/w$)) and subacute toxicity (maximum 1 g/kg BW) in the mice, being safe for consumption [21]. The authors also supplemented C-PC at 100, 200, 500, and 1000 mg/kg BW and reported the immunomodulatory potential of C-PC in suppressing the synthesis of pro-inflammatory cytokines, interferon-$\gamma$ (INF-$\gamma$), and tumor necrosis factor-$\alpha$ (TNF-$\alpha$) in a dose-dependent manner. Doses of 500 and 1000 mg/kg significantly decreased TNF-$\alpha$ and also enhanced the serum antioxidant enzyme activity. Doses of 200, 500, and 1000 mg/kg reduced considerably (INF-$\gamma$) and significantly increased the anti-inflammatory interleukin-10 (IL-10). No significant differences were observed in BW, neither organ weight, hematological red and white parameters, and biochemical values, including bilirubin, total protein, albumin, globulin, total cholesterol, serum glutamic-oxaloacetic transaminase, serum glutamate pyruvate transaminase, among others. In this study [21], the C-PC extraction method was not described.

There are still challenges for C-PC application primarily due to the extraction methods resulting in low purity extracts and low stability of the pigment under storage and food process [4]. At the moment, combining C-PC and sugars, such as sucrose, improved its

stability due to modifications in molecular interactions [16], which is an advantage for food matrices, likewise, the diet provided for Swiss mice in this study. Additionally, given that after C-PC extraction, the RB preserves proteins, carbohydrates, polyunsaturated fatty acids such as α-linolenic acid, minerals, carotenoids, and tocopherol [22], being nutritionally appealing instead of simply discarded as waste. The idea of complete utilizing the *Spirulina* biomass and the C-PC extraction mediated by greener processes propelled this study that aimed to evaluate different parameters in the green process of organic *Spirulina* biomass C-PC extraction to understand the impact on weight and glucose tolerance of C-PC extract, SB and RB in Swiss mice fed a high-fat diet (HFD). The potential of this study is also revealed by the detailed description of the green extraction methods of the inputs applied and their antioxidant activity.

## 2. Material and Methods

### 2.1. Material

*Arthrospira platensis*, popularly known as *Spirulina*, was the cyanobacteria selected in this research. Different solvents were used for each step of this research: distilled water (in the proportion of 0.16: 1 of biomass to solvent ratio) for C-PC extraction, solid ammonium sulfate, and sodium phosphate buffer (pH 7.0) for partial purification. Petroleum ether for fat content and sulfuric acid, catalytic mixture, sodium hydroxide, and boric acid for protein content determination; all the solvents were used in analytical grades. For the antioxidant methods, acetone, methyl alcohol, DPPH (2,2-Diphenyl-1-picryl-hydrazyl), 2,2′-azobis(2-amidinopropane) dihydrochloride, fluorescein, phosphate buffer, ABTS•+ radical, and potassium persulfate were also used.

### 2.2. Spirulina Biomass Selection

Before selecting and purchasing *Spirulina* biomasses, a broad search was conducted in health food stores in São Paulo state (Brazil) and e-commerce. Three different *Spirulina* biomass were tested to extract C-PC. Among them, three were used in the present work, B.1—organic powdered biomass—Fazenda Tamanduá® (Santa Teresinha, Brazil) (SFT); B.2—conventional granulated biomass—the Federal University of Goiás in partnership with Brasil Vittal® (Brasília, Brazil) company (SBV), and B.3—conventional powdered biomass—Sunrise Nutrachem Group® (Qingdao, China) (SNG) (Figure 1).

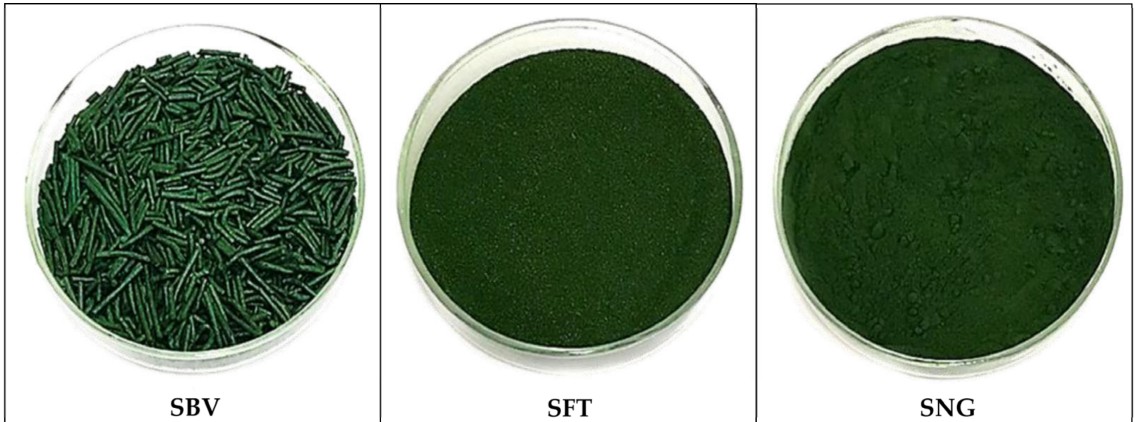

**Figure 1.** *Spirulina* biomass samples. Subtitle: SBV—granulated *Spirulina* biomass—the Federal University of Goiás in partnership with Brasil Vittal® (Brasília, Brazil) company; SFT—organic powdered *Spirulina* biomass—Fazenda Tamanduá® (Santa Teresinha, Brazil); SNG—powdered *Spirulina* biomass—Sunrise Nutrachem Group® (Qingdao, China).

Initial tests were performed with these three distinct biomasses purchased to obtain the one with the highest concentrations, purity, and yield of C-PC, as well as the best economic feasibility, environmental, and social value.

### 2.3. Solid-Liquid Watery Extraction (SLWE) of C-PC

*Spirulina* biomass was removed from the freezer (−38 °C—Liobras® FV500, São Carlos, Brazil) 1 h before the extraction and was processed in an analytical mill (IKA A11 Basic) to disrupt the cells. To separate the particles and promote the standardization of *Spirulina* biomass, an inox test sieve (ASTM E11—140 MESH/Tyler 150, Gilson Company, Lewis Center, OH, USA) was used, and the extraction was based on the methodology described by Moraes et al. [5].

C-PC extraction from *Spirulina* dry biomass was carried out by adding distilled water (50 mL) to the *Spirulina* different samples (8 g of SFT, SBV, or SNG) into an Erlenmeyer flask. The solid-liquid extraction was made in triplicate at room temperature and protected from light for 60 min. The mixture was centrifuged (NT 816, Nova Analítica®, São Paulo, Brazil) under refrigeration (10 °C) for 30 min at 10,000× *g*. The supernatant was collected and separated from the remaining biomass (RB). The centrifugation and separation process were repeated three more times.

The extracts were vacuum filtered using qualitative filter paper (Unifil®, Brazil), and the solution was twice filtered. For the retention of hydrophilic compounds, a 0.22 µm hydrophilic syringe filter (Analítica®, São Paulo, Brazil) was used (C-PC SM). Tests were also made without any filtration step (C-PC NF). The supernatant (C-PC) and precipitated (RB) were stored in the freezer (Liobras® FV500, São Paulo, Brazil) at −38 °C for subsequent analysis of the C-PC standardized method (C-PC SM); this entire process can be seen in Figure 2.

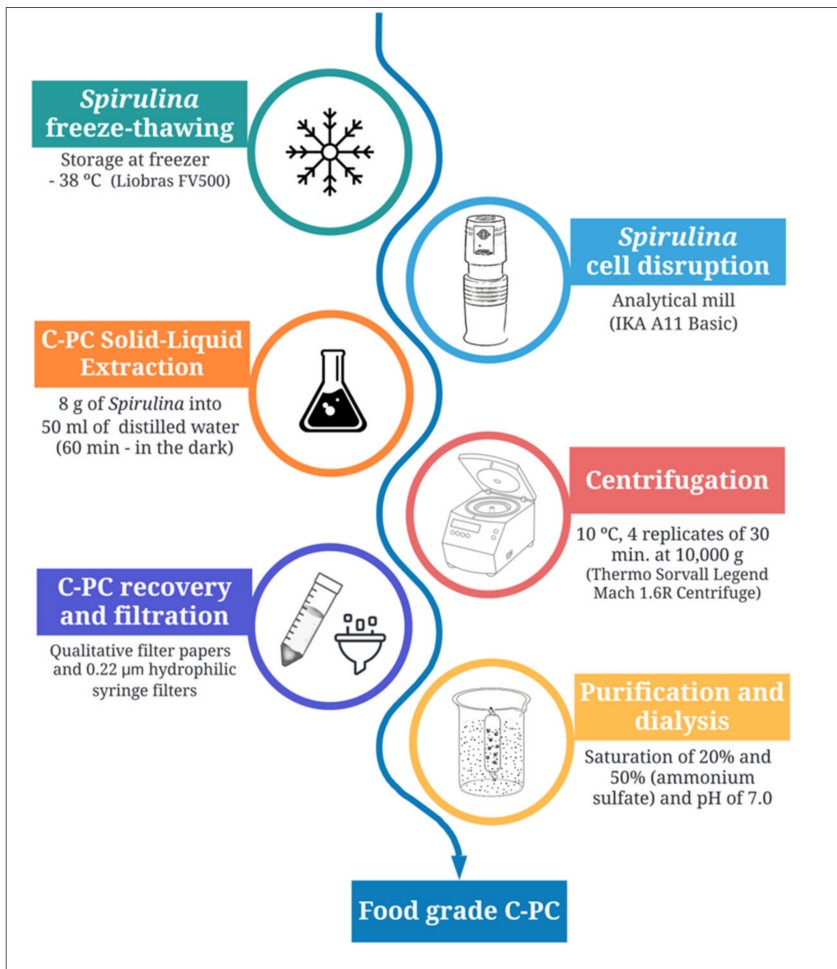

**Figure 2.** C-phycocyanin (C-PC) solid-liquid extraction process adapted from Moraes et al. [5].

### 2.4. C-phycocyanin Purification by Fractional Precipitation with Ammonium Sulfate

Fractional precipitation with ammonium sulfate $(NH_4)_2 SO_4$ was performed according to Amarante et al., [1] adapted from Silva et al. [23], Moraes et al. [5], and Figueira [24]. In the first fractionation, solid $(NH_4)_2 SO_4$ was added to C-PC clarification extract at a saturation concentration equivalent to 20%, under constant stirring for 30 min, followed by incubation at 4 °C for 2 h. After this period, the sample was centrifuged (NT 816, Nova Analítica®, São Paulo, Brazil, $12,000 \times g$, 10 °C, 20 min), and the precipitate was discarded. In the second fractionation, solid $(NH_4)_2 SO_4$ was added to the supernatant from the previous step until 50% saturation. The solution was maintained in the same conditions as the first fractionation and then centrifuged. The supernatant was discarded, and the precipitate was resuspended in 0.05 mol/L sodium phosphate buffer pH 7.0 at a resuspension volume/initial volume ratio of 0.52. The resolubilized precipitates were dialyzed using 0.05 mol/L sodium phosphate buffer pH 7.0 for salt removal (Figure 2).

Before and after the partial purification process, analytical methods were performed for the calculation of C-PC concentration in mg/mL (Equation (1)), purity (Equation (2)), and Yield in mg/g (Equation (3)), using a UV-visible spectrophotometer (Varian Cary® 50 Bio). For better comprehension, it is essential to know about the following equations where A corresponds to Absorbance readings at the wavelengths of 620 nm (absolute absorption peak of C-PC), 652 nm (absorption maximum of Allophycocyanin), and 280 nm (total proteins). In Equation (1), Fdil is the dilution factor. In Equation (3), V corresponds to solvent volume (mL), and DB is the quantity of dry biomass (g).

$$C - PC\ Concentration\ (mg/mL) = \frac{A_{620} - 0.474 \times A_{652}}{5.34} \times Fdil \tag{1}$$

$$C - PC\ Purity = \frac{A_{620}}{A_{280}} \tag{2}$$

$$C - PC\ Yield\ (mg/g) = \frac{C - PC\ concentration \times V}{DB} \tag{3}$$

$$C - PC\ Yield\ after\ partial\ purification\ (\%) = \frac{\text{C-PC final concentration} \times \text{Final Volume}}{\text{C-PC initial concentration} \times \text{Initial Volume}} \times 100 \tag{4}$$

### 2.5. Proximate Composition of Spirulina Biomass (SB), Remaining Biomass (RB), and Partial Purified C-Phycocyanin (C-PC)

The proximate composition of the *Spirulina* organic powdered biomass—Fazenda Tamanduá® (Santa Teresinha, Brazil) (SFT) and the remaining biomass (RB) after C-PC extraction were determined using AOAC methods in the dry samples, except for moisture [25]. The forced air oven was used to establish the sample's moisture. It was highlighted that the SB moisture content was provided by Tamanduá® (Santa Teresinha, Brazil) technical regulation (at 50 °C for 12 h). The RB and C-PC moisture content was obtained at 75 °C for 4 and 5 h, respectively. The ash content was based on incineration in a muffle furnace at 550 °C (Chapter 16—AOAC, 2005). The total fat content was determined by Soxhlet extraction (Chapter 17—AOAC, 2005). The protein content was determined by the total nitrogen obtained via micro-Kjeldahl and considering the conversion factor% N × 6.25 (Chapter 18—AOAC, 2005). Carbohydrate was done by calculation: Carbohydrates (g) = 100 − (moisture + ash + total fat and protein). In general, data were expressed as g/100 g on a wet basis, which was attainment by conversion calculation: Wet basis (%) = (value on dry basis ∗ total solids)/100, the protein content was also described on the dry basis to facilitate the literature results comparison. Reinforcing that the ash, total fat, and protein analysis was performed only in SB and RB samples and the moisture in C-PC, SB, and RB samples. All the analyses were made in triplicate. The protein concentration was measured according to the method of Bradford [26] (Bio-Rad, CA, USA).

*2.6. Antioxidant Activity of C-PC*

The samples were extracted with 20 mL of methanol using a magnetic stirrer for 15 min to extract the antioxidant compounds from the matrices. Samples were filtered, and the retentate was washed twice with an additional 10 mL of methanol in each. The filtrate was then concentrated using a rotary evaporator at temperatures around 37 °C [1,27], and the samples were filtered using 0.22 μm cellulose acetate membranes (Analítica®, São Paulo, Brazil) before analysis.

For the determination of the antioxidant activity against the peroxyl radical, the ORAC (oxygen radical absorption capacity) method was used, based on peroxyl radical formation by the thermal degradation of AAPH (2,2′-azobis (2-amidinopropane) dihydrochloride) at 37 °C. One hundred and fifty microliters of fluorescein (61 mmol/L prepared in 75 mmol/L phosphate buffer pH 7.4) were added to 96-well microplates, followed by 25 μL of the extract made (diluted 100, 500, and 1000 times in phosphate buffer) or buffer or Trolox standard solution (50 μmol/L). The plate was incubated for 10 min at 37 °C under intermittent stirring, and after that, 25 μL of AAPH solution (19 mmol/L, prepared in phosphate buffer) was added to each well. The fluorescence readings at 538 nm (excitation at 485 nm) are performed every minute for 180 minutes at Spectramax® (Thermo Fisher Scientific, MA, USA) [28].

For the determination of the antioxidant activity against the ABTS radical, 30 μL of the extracts were added to 3 mL of the ABTS•+ radical solution, and the absorbance of the reaction was measured in the spectrophotometer (Varian Cary® 50 Bio) at 734 nm after 6 min. The ABTS•+ radical solution was prepared from the reaction between 5 mL of stock solution of ABTS (7 mmol/L) and 88 μL of potassium persulfate (140 mmol/L), formulated in distilled water. The mixture was kept in the dark at room temperature for 16 h. This solution was diluted with ethanol to an absorbance of 0.8 ± 0.05 at 734 nm. A standard Trolox curve in ethanol, constructed under the same reaction conditions, is used to quantify the total antioxidant activity [29]. Both antioxidant results were expressed in μ mol/g sample of Trolox equivalents (TE) and were made in triplicate.

The antioxidant activity methodology against DPPH (2,2-difenil-1-picril-hidrazil) was adapted from Larrauri et al. [30]. An aliquot of 0.1 mL of each was obtained from the extracts, transferred to test tubes with 3.9 mL of DPPH radical, and homogenized on a shaker. The absorbance measurement was monitored every minute in the spectrophotometer (Varian Cary® 50 Bio, Thermo Fisher Scientific, MA, USA) at 515 nm.

*2.7. Design and Animals' Experimental Procedures*

The Experimental Research Committee of the Federal University of São Paulo (UNIFESP) approved all procedures for the care of the animals used in this study (CEUA-UNIFESP N° 8626090821). Thirty Swiss mice were provided from Campinas State University (Universidade Estadual de Campinas—CEMIB, São Paulo, Brazil), kept under controlled conditions of 12 h of light (lights on at 6 a.m.) and 12 h of the dark cycle (lights off at 6 p.m.), and temperature (24 ± 1 °C) at 60 ± 5% humidity. The animals had free access to food and water, and the supplementation was made by intragastric gavage.

Swiss mice were distributed semi-randomly into the following five groups (with 5–6 animals for each treatment): 1. The control diet (CD): normolipidic diet, receiving 0.09% of saline solution by intragastric gavage; 2. High Fat Diet (HFD): hyperlipidic diet, receiving 0.09% of saline solution by intragastric gavage; 3. High Fat Diet plus *Spirulina* (HFDS): hyperlipidic diet, receiving 500 mg/kg of body weight per day of SB by intragastric gavage; 4. High Fat Diet plus C-PC (HFDC): hyperlipidic diet, receiving 500 mg/kg of body weight per day of C-PC rich extract, by intragastric gavage, without chemicals, only water as a solvent, as a greener approach, and 5. High Fat Diet plus remaining biomass (HFDR): hyperlipidic diet, receiving 500 mg/kg of body weight per day of RB, precipitate after C-PC extraction by intragastric gavage. The control diet (CD) was prepared according to the recommendations of the American Institute of Nutrition (AIN-93) [31]. The High-Fat Diet (HFD) was adapted from CD, and its composition can be seen in Table 1. C-PC extract

and RB were freeze-dried to powder using a lyophilizer (Liobras® L101). All treatments were performed using the same lot. It is crucial to notice that the supplementation amount standardization was dependent on Swiss mice weight, that is, 500 mg of SB, C-PC, or RB per each kg of animal weight. The C-PC, RB, and SB were dissolved in water in Swiss mice's intragastric gavage.

**Table 1.** Nutritional value in 100 grams (g) of Control diet (CD) and High-Fat Diet (HFD).

| Ingredients | CD [32] | HFD Adapted from [31,32] |
| --- | --- | --- |
| Cornstarch (g) | 72.07 | 45.00 |
| Sucrose (g) | — | 15.00 |
| Casein (g) | 14.00 | 18.00 |
| Soybean oil (g) | 4.00 | 4.00 |
| Lard (g) | — | 18.00 |
| Cellulose (g) | 5.00 | — |
| Vitamin Mix (g) | 1.00 | 1.00 |
| Mineral Mix (g) | 3.50 | 3.50 |
| L-cystine (g) | 0.18 | 0.18 |
| Choline bitartrate (g) | 0.25 | 0.25 |
| BHT (preservative and antioxidant agent) (g) | 0.0008 | 0.0008 |
| Total energy (Kcal) | 380.28 | 510.00 |

After 12 weeks of treatment, an oral glucose tolerance test (OGTT) was performed from 1 to 4 p.m. After a 4 h time-restricted feeding, basal glycemia was measured from the tail vein using a glucometer (Accu-Check, Roche Diagnóstica Brasil Ltd.a., São Paulo, SP, Brazil). Immediately after the first measure, glucose solution (in a concentration of 1 g/kg) was administrated by intragastric gavage. Blood samples were collected after 15, 30, 45, 60, and 120 min to obtain the area under the glycemic curve (AUC). The food intake was monitored daily, and the animals were weighed weekly.

At the end of the experimental period, seven days after 12-week OGTT, animals fasted for 12 h overnight before being euthanatized by a lethal dose of anesthetic (80 mg/kg body weight of ketamine and 10 mg/kg body weight of xylazine).

*2.8. Statistical Analysis*

First, distribution assumptions were verified by the Shapiro–Wilk, Kolmogorov–Smirnov, and Chi-Square tests. The homogeneity tests, by the Cochran C, Hartley, Bartlett test, when $p > 0.05$, were classified as parametric. Therefore, the measurements from the assays were carried out independently in triplicate and compared by applying one-way analysis of variance (ANOVA) with Tukey's post hoc test. All data were expressed as mean ± standard deviation. The considered degree of significance was 95% ($p \leq 0.05$). Statistical analyses were performed using Microsoft® Excel® (version 2206 64-bit, Redmond, WA, USA) and software STATISTICA® (version 13.5.0.17 64-bit, Rome, Italy). The calculation of percentage change was demonstrated in the text: % change = (higher value − lower value/lower value) ∗ 100, to demonstrate the increment, in percentual (%), between two values.

**3. Results**

*3.1. Comparison between Conventional and Organic Spirulina Biomass Using SLWE*

For the commercial *Spirulina* biomass selection (SBV, SFT, and SNG) with the best results of concentration, purity, and yield after C-PC extraction (Equations (1) to (3)), initial tests were performed following Moraes et al. [5] method (Table 2). The parameters of SNG were the lowest in comparison with SBV and SFT, and incompatible with Moraes et al. [5] methodology study that showed in C-PC solid-liquid water extraction, C-PC concentration, purity, and yield of 13.20 mg/mL, 0.603 and 82.48 mg/g, respectively. Therefore, SNG

biomass was the first to be excluded for the continuity of the study because of the worst C-PC analytical parameters.

**Table 2.** Comparison of C-PC concentration, purity, and yield results using the method of Moraes et al. [5] to understand the different parameters of three commercial *Spirulina* biomass.

|  | SBV | SFT | SNG |
|---|---|---|---|
| **Concentration (mg/mL)** | 5.34 [a] ± 0.56 | 2.83 [b] ± 0.69 | 0.60 [c] ± 0.27 |
| **Purity** | 0.35 [a] ± 0.02 | 0.22 [b] ± 0.01 | 0.06 [c] ± 0.02 |
| **Yield (mg/g)** | 33.39 [a] ± 3.49 | 17.69 [b] ± 4.30 | 3.73 [c] ± 1.71 |

Different letters in the same line represent different values for each one since $p > 0.05$ in the analysis of variance (Tukey's test). Values are mean ± standard deviation. SBV = conventional granulated biomass sample from the Federal University of Goiás in partnership with Brasil Vittal® company. SFT = organic powdered biomass sample from Fazenda Tamanduá®. SNG = conventional powdered biomass sample from Sunrise Nutrachem Group®.

Compared to SFT, the SBV had an increment of percentage change in concentration, yield, and purity of 88.69%, 59.09%, and 88.75%, concomitantly. Still, both SFT and SBV presented results smaller than those described by Moraes et al. [5]. The first steps to C-PC extraction improvement are *Spirulina* cell disruption to C-PC releasement and samples' homogenization; these two were twice as fast in powder samples (SFT and SNG) than in granulated biomass (SBV), which promoted energy savings and can also improve the operational team performance. Even though the best C-PC parameters can be seen in SBV biomass, SFT organic biomass was chosen for the sequence of the experiments, taking into consideration not only the analytical parameters but also the facility of C-PC extraction, fastest time, economic feasibility, environmental and social value, being under the principles of green chemistry [14].

To understand how each extraction step influenced the parameters of *Spirulina* extract (C-PC), Table 3 shows the comparison of C-PC SM (standardized methodology), C-PC NF (no filtration), and C-PC purified. The purification by fractional precipitation with $NH_4)_2 SO_4$ as described by Figueira et al. [24] is beneficial to remove the contaminants of the C-PC sample, among them, *Spirulina* cellular matrix components as chlorophyll [33] and proteins [16] that can be available during the manufacturing process.

**Table 3.** Comparison of C-PC analytical parameters for extraction process improvement based on standardized Moraes et al. [5] method.

|  | C-PC SM | C-PC NF | C-PC Purified |
|---|---|---|---|
| **Concentration (mg/mL)** | 5.63 [a] ± 0.52 | 5.10 [a] ± 0.46 | 4.50 [a] ± 0.73 |
| **Purity** | 0.38 [b] ± 0.06 | 0.27 [b] ± 0.03 | 0.81 [a] ± 0.11 |
| **Yield (mg/g)** | 35.33 [a] ± 3.43 | 31.86 [a] ± 2.90 | * |

Different letters in the same line represent different values for each one since $p > 0.05$ in the analysis of variance (Tukey's test). Values are mean ± standard deviation. (*) Following Equation (4)—C-PC Purified Yield was (%): 89.51 ± 7.00. C-PC SM = C-PC Extraction Standardized Methodology adapted from Moraes et al., 2010, without partial purification. C-PC NF = C-PC Extraction no filtration step. C-PC Purified = C-PC Extraction Standardized Methodology adapted from Moraes et al., 2010, with partial purification.

Thus, in comparison with C-PC SM and C-PC NF, which showed similar results in all the parameters, the C-PC Purified sample showed an increment in purity ratio of 53.1% and 66.7%, respectively, demonstrating a food grade purity (>0.70). Thereby this confirms that the purification step is critical, even because it is related to product cost; that is, the purer the C-PC (analytical grade purity > 4.0), the higher the price and manufacturing costs in the cosmetic, agrochemistry, and food industries for USD 60.00 per gram [34], and in C-PC with analytical grade, the cost can reach USD 19,500 per gram [35]. The C-PC price has increased in recent years in comparison with previous studies as described by Figueira et al. [24] at the cost of approximately USD 0.35 per gram as a C-PC biocolorant [36] and USD 4500.00 per gram of C-PC in analytical grade [37].

### 3.2. Proximate Composition and Antioxidant Activity Effects of SB, RB, and Partial Purified C-Phycocyanin (C-PC)

The proximate composition (Table 4) revealed that RB contains a higher percentage change than SB of ash (in 256.25%) and total fat (in 67.85%). The data collected at *Spirulina* biomass label from Fazenda Tamanduá® (SB FTL, Santa Teresinha, Brazil) provided slight differences in grams compared to the composition of SB: 0.14 g of ash, 1.05 of total fat, and 0.26 g of protein. The moisture content in three samples was about 80%, and the moisture value of SB FTL was used for SB since the sample came freeze-dried from the Fazenda Tamanduá® company. Since C-PC is a phycobiliprotein in terms of composition, only the total protein was determined from the purified extract using the method described by Bradford [26], resulting in 118.97 ± 16.09 µg/g of protein in dry biomass. Additionally, it is essential to highlight that the moisture of C-PC extract before and after purification was 93.02% (±0.04) and 97.63% (±0.02), respectively.

**Table 4.** Proximate composition on the wet basis of *Spirulina* biomass (SB) and remaining biomass (RB) after C-PC extraction (g/100 g).

|  | SB | RB | SB FTL [2] |
|---|---|---|---|
| **Moisture** | 80.00 [1] | 82.22 ± 0.40 | 80.00 [1] |
| **Ash** | 0.16 ± 0.00 | 0.57 ± 0.01 | 0.30 |
| **Total fat** | 0.28 ± 0.01 | 0.47 ± 0.18 | 1.33 |
| **Protein** | 10.41 ± 0.04 | 10.87 ± 0.62 | 10.67 |
| **Carbohydrate** | 9.15 ± 0.04 | 5.83 ± 0.41 | 7.70 |

Values are mean ± standard deviation. [1] Moisture data was provided by Tamanduá® technical regulation. [2] SB FTL = *Spirulina* biomass from Fazenda Tamanduá® Label data. SB = *Spirulina* biomass from Fazenda Tamanduá®. RB = Remaining biomass after C-PC extraction.

The antioxidant activity was determined to facilitate the comprehension of each component (C-PC, SB, and RB) used for the in vivo experiments. The mean values of C-PC and SB did not show statistically significant differences, presenting higher values than RB in ORAC, ABTS, and DPPH results (Table 5).

**Table 5.** Antioxidant activity of C-PC, *Spirulina* biomass (SB), and remaining biomass (RB) was determined by ORAC, ABTS, and DPPH methods.

| Antioxidant Method | C-PC | SB | RB |
|---|---|---|---|
| ORAC (µM TE/g) | 371.8 [a] ± 54.2 | 359.2 [a] ± 39.3 | 132.5 [b] ± 41.0 |
| ABTS (µM TE/g) | 10.06 [a] ± 1.12 | 9.54 [a] ± 2.70 | 5.14 [b] ± 1.70 |
| DPPH (EC$_{50}$ g/g) | 501.8 [a] ± 51.1 | 478.5 [a] ± 31.5 | 191.4 [b] ± 2.6 |

Values are mean ± standard deviation. Different letters in the same line represent different values for each one since $p > 0.05$ in the analysis of variance (Tukey's test). Values are mean ± standard deviation.

### 3.3. Biological Effects of SB, C-PC, or RB in Swiss Mice Fed with HFD

Considering the weeks evaluated and described in Table 6, the highest dietary intake of Swiss mice in the control group (CD) can be observed. Additionally, in weeks 3, 5, 6, 7, 8, 9, and 12, all the other groups (HFD, HFDS, HFDC, and HFDR) had statistically the same dietary intake in grams. According to these data, it can be assumed that the consumption differences seemed to be related to the type of diet (CD—normolipidic or HFD—high-fat diet) and not to the added components (*Spirulina*, C-PC, or remaining biomass) during the experiments.

**Table 6.** Dietary intake in grams (g) of Swiss mice per week in different treatments.

| Groups | Weeks of Treatment | | | | | | | | | | | |
|---|---|---|---|---|---|---|---|---|---|---|---|---|
| | 1 | 2 | 3 | 4 | 5 | 6 | 7 | 8 | 9 | 10 | 11 | 12 |
| CD (g) | 55.7 [a] ± 2.2 | 59.8 [a] ± 0.6 | 58.1 [a] ± 5.1 | 44.6 [a] ± 1.8 | 68.1 [a] ± 7.4 | 59.2 [a] ± 5.7 | 69.7 [a] ± 8.5 | 51.3 [a] ± 10.9 | 73.5 [a] ± 7.5 | 80.0 [a] ± 1.0 | 87.4 [a] ± 9.1 | 55.5 [a] ± 0.2 |
| HFD (g) | 36.6 [bc] ± 0.9 | 39.8 [c] ± 4.2 | 20.6 [b] ± 2.1 | 27.0 [bc] ± 1.4 | 28.3 [b] ± 1.3 | 27.6 [b] ± 1.4 | 26.8 [b] ± 0.3 | 23.5 [b] ± 1.3 | 35.7 [b] ± 4.5 | 35.0 [c] ± 5.8 | 20.2 [c] ± 8.5 | 23.6 [b] ± 0.9 |
| HFDS (g) | 36.5 [bc] ± 2.5 | 40.6 [b] ± 0.3 | 17.5 [b] ± 0.3 | 29.1 [bc] ± 1.0 | 29.8 [b] ± 1.7 | 29.4 [b] ± 1.8 | 29.2 [b] ± 1.2 | 26.5 [b] ± 2.2 | 28.7 [b] ± 3.2 | 48.8 [b] ± 0.6 | 28.3 [bc] ± 3.1 | 25.2 [b] ± 2.8 |
| HFDC (g) | 34.7 [bc] ± 2.4 | 44.8 [bc] ± 0.4 | 18.5 [b] ± 2.8 | 26.5 [c] ± 0.7 | 28.0 [b] ± 0.2 | 25.3 [b] ± 0.6 | 28.3 [b] ± 0.1 | 25.3 [b] ± 0.3 | 30.0 [b] ± 0.3 | 46.6 [b] ± 0.4 | 28.7 [b] ± 0.3 | 25.3 [b] ± 0.5 |
| HFDR (g) | 41.5 [b] ± 2.1 | 50.4 [b] ± 2.8 | 23.2 [b] ± 2.6 | 32.7 [b] ± 4.3 | 36.0 [b] ± 2.8 | 41.7 [b] ± 12.8 | 36.2 [b] ± 6.0 | 29.7 [b] ± 5.5 | 40.8 [b] ± 8.4 | 49.4 [b] ± 5.1 | 41.1 [b] ± 11.6 | 33.0 [b] ± 8.3 |

Different letters in the same column represent different values for each one since *p* > 0.05 in the analysis of variance (Tukey's test). Values are mean ± standard deviation.

　　　　Regarding the Δ weight (Table 7), it could be observed that the highest results were related to *Spirulina* (HFDS) and remaining biomass (HFDR) without significant differences among them. Focusing on the C-PC group (HFDC), the lowest values of Δ weight, which are equal to CD and HFD, are also HFDR treatments. In summary, HFDC, CD, and HFD are significantly equal and differ only from the HFDS treatment. At the final weight after treatments, HFDS, HFDR, and HFD groups showed the highest weight values, and HFDS and HFDR differed from CD and HFDC. These differences cannot be seen in the initial weight, where the HFDS group has the lowest weight values, similar to HFD, HFDC, and HFDR. Table 8 shows the Swiss mice tissue weight and liver weight differences between the treatments.

**Table 7.** Swiss mice weight in grams (g) before and after treatments and delta (Δ) weight (g) before treatment.

| Groups | Initial Weight [1] | Final Weight after Treatment [2] | Δ Weight [3] |
|---|---|---|---|
| **CD** | 27.8 [a] ± 4.0 | 49.1 [b] ± 5.0 | 14.2 [b] ± 5.0 |
| **HFD** | 23.8 [ab] ± 1.5 | 52.8 [ab] ± 2.9 | 14.8 [b] ± 5.2 |
| **HFDS** | 23.0 [b] ± 2.3 | 59.6 [a] ± 3.7 | 24.1 [a] ± 3.2 |
| **HFDC** | 24.4 [ab] ± 2.2 | 48.3 [b] ± 3.4 | 12.7 [b] ± 3.4 |
| **HFDR** | 24.3 [ab] ± 1.0 | 57.3 [a] ± 3.6 | 19.0 [ab] ± 4.2 |

Different letters in the same column represent different values for each one since *p* > 0.05 in the analysis of variance (Tukey's test). Values are mean ± standard deviation. [1] Initial Weight = animals' weight before the beginning of treatment. [2] Final Weight after treatment = animals' weight after 12 weeks of treatment. [3] Weight = final weight after treatment (week 12)—initial weight after treatment (week 1).

**Table 8.** Swiss mice tissue weight and liver in grams (g) after treatments.

| Groups | Epididymal (EPI) | Mesenteric (MES) | Inguinal (ING) | Retroperitoneal (RET) | Liver (LIV) |
|---|---|---|---|---|---|
| **CD** | 2.0 [a] ± 0.8 | 1.1 [a] ± 0.5 | 1.0 [b] ± 0.5 | 0.4 [a] ± 0.1 | 1.9 [b] ± 0.2 |
| **HFD** | 2.1 [a] ± 0.7 | 1.3 [a] ± 0.2 | 1.6 [ab] ± 0.4 | 0.5 [a] ± 0.1 | 2.3 [ab] ± 0.5 |
| **HFDS** | 2.2 [a] ± 0.5 | 1.5 [a] ± 0.3 | 2.2 [a] ± 0.7 | 0.6 [a] ± 0.2 | 2.9 [ab] ± 0.6 |
| **HFDC** | 1.6 [a] ± 0.4 | 0.9 [a] ± 0.5 | 1.1 [b] ± 0.5 | 0.4 [a] ± 0.1 | 1.9 [b] ± 0.1 |
| **HFDR** | 1.8 [a] ± 0.5 | 1.4 [a] ± 0.3 | 1.8 [a] ± 0.5 | 0.4 [a] ± 0.1 | 3.5 [a] ± 1.8 |

Different letters in the same column represent different values for each one since *p* > 0.05 in the analysis of variance (Tukey's test). Values are mean ± standard deviation.

　　　　In Table 8, the visceral adipose tissue represented by epididymal (EPI), mesenteric (MES), and retroperitoneal (RET) did not show differences between treatments with or without SB, C-PC, or RB supplementation and HFD. Additionally, white adipose tissue can be demonstrated by the EPI, MES, inguinal (ING), and RET tissues. Differences just can be seen in ING tissue, indicating that the supplementation with C-PC (HFDC) helped with the lowest values in comparison with HFDS and HFDR; with the ING highest weight tissue values, further analysis needs to be conducted to understand C-PC biological mechanisms in mice.

### 3.4. Repercussions of SB, C-PC, or RB on Glucose Metabolism of Swiss Mice HFD Fed

Figure 3 shows the behavior of Swiss mice fasting blood-glucose average after 15, 30, 60, 90, and 120 min of glucose [1 g/kg] intragastrically gavaged. The glycemia average of CD and HFDC were lower than the others. Additionally, in the first 15 min, the AUC of all treatments had the most prominent peak and after 60 min showed an interesting decrease in AUC of 8-week and 12-week treatments. In AUC 8-week treatments, the glycemia after 120 min went back to baseline (0 min), but in the AUC 12-week treatment, just in the CD group, the glycemia average returned to the baseline as expected [38].

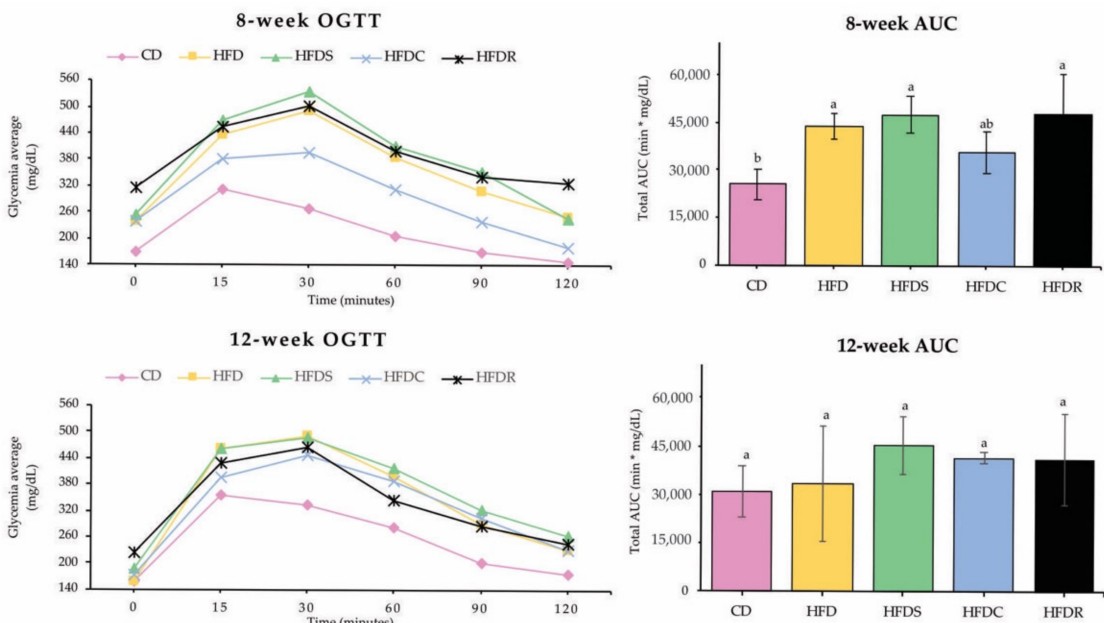

**Figure 3.** Oral glucose tolerance test (OGTT) and area under the curve (AUC) of Swiss mice animals in different groups: CD (Control diet), HFD (High Fat Diet), HFDS (High Fat Diet plus *Spirulina*), HFDC (High Fat Diet plus C-PC) and HFDR (High Fat Diet plus remaining biomass) after 8 and 12 weeks after treatment. The bars in AUC graphics represent the standard deviation, and the letters above indicate differences between groups ($p \leq 0.05$).

In addition, Figure 3 showed that after eight weeks of treatment, the group CD presented a lower area under the curve (AUC) than HFD, HFDS, and HFDR, resembling only the HFDC group ($p \leq 0.05$). However, after twelve weeks, the AUC results are equalized, and the differences between treatments disappear.

## 4. Discussion

Many studies have determined the best methodologies for C-PC extraction from *Spirulina* with suitable analytical parameters [16]. However, few studies proposed to determine a C-PC extended comprehension, since the extraction and partial purification until the behavior of C-PC in vivo knowledge, as well as understanding physicochemical analyses and antioxidant activity [14], as this research.

The most widely used methods for C-PC extraction over the last ten years are Ultrasound-Assisted Extraction (UAE), Microwave Assisted Extraction (MAE), Freeze-Thawing (FT), Aqueous Two-Phase System or Aqueous Two-Phase Extraction (ATPS/ATPE), and Pulsed Electric Field (PEF), as well as Solid-Liquid Extraction (SLE) that was used in this research [14]. All these techniques have been the essential tools for greening the C-PC extraction methods when it emphasizes the utilization of green reagents—as ionic liquids (ILs) [39], natural deep eutectic solvents (DES) [40], and water, the solvent used in this study, as well as the minimization, automation, and the reduction in time taken and op-



erator effort [41], that which makes these methods commonly linked to Green Analytical Chemistry (GAC).

The selection of SFT biomass with faster cell disruption than SBV and the utilization of remaining biomass (RB) before and after the C-PC extraction process demonstrated the researchers' efforts and new hypotheses' propositions of economic and environmental sustainability from *Spirulina* biomass bioproducts. Further analysis is needed for a better comprehension of life cycle assessment (LCA) improvement [42].

Despite this report being based on the C-PC extraction methodology from Moraes et al. [5] with the same biomass and water solvent ratio (0.16:1), differences in C-PC analytical parameters (concentration, purity, and yield) were seen between them. These variations can be explained by disparities in operators, equipment, and raw material type (*Spirulina* biomass), as seen in Table 2, complicating the study results' reproducibility.

To support these findings, Ílter et al. [43] used various C-PC extraction methods (classical homogenization, MAE, and UAE) with differences in solvents and *Spirulina* biomass physical forms (wet, dried, and frozen) and presented too many variations in results between them. The optimized condition was obtained with UAE, even without a vivid blue color as seen in classical homogenization, and the authors assume that a highly C-PC selective downstream extraction process is more effective in terms of high extraction yield and C-PC concentration, time, and cost than the vivid blue color. Therefore, the physical form of the biomass, the type of solvent, extraction methods, and the level of their process variables should be determined considering the targets in the extraction process [43].

Furthermore, as described by Figueira et al. [24], to remove sample contaminants, a purification process is necessary (Table 3) to produce higher C-PC extract purity, even though it is an onerous experimental procedure and is also time-consuming which comprehends multiple complex steps. Therefore, purity is responsible for the major differences in C-PC prices due to the obstacles in its purification process [4].

To natural blue C-PC color preservation and C-PC stability during extraction, it is necessary to facilitate the release of C-PC, but not other impurities; thus, the chemical C-PC extraction (using phosphate buffer 100 mM and pH 7.5, for 3 h at room temperature and 5 g/L NaCl, with pH adjustment after extraction to 6.0–6.5) seemed to facilitate the immediate release of C-PC from dried biomass, when compared to other methodologies, that can prove that the best method will depend on several factors, and the protocol may be better for one than another variable of interest.

Amarante et al. [1] study demonstrated that C-PC purified extract by ultrafiltration with a 50 kDa polyethersulfone membrane followed by ion exchange chromatography with pH gradient elution resulted in a C-PC purity of 3.90 and a concentration of 0.91 mg/mL, and this methodology has enabled a positive impact in comparison of C-PC clarified extract (purity of 0.90 and concentration of 0.09 mg/mL) in the antioxidant activity with an increment of 18.27-fold against the ABTS and 40.65-fold peroxyl ORAC radicals, respectively. The C-PC content in Amarante et al. [1] demonstrated a higher percentage change in purity (11.11% in C-PC clarified and 381.48% in C-PC purified) than in C-PC purified in the current study. In addition, Amarante et al. [1] showed higher antioxidant activity and C-PC concentration results in C-PC purified extracts representing 161.66 μM TE/g in ABTS, 1211.41 μM TE/g in ORAC, and 0.908 mg/mL, respectively, when compared with C-PC purified in this paper's findings.

Although the clarified C-PC in Amarante et al. [1] showed lower values against ABTS (8.85 μM TE/g) and ORAC (29.80 μM TE/g) radicals and in C-PC concentration (8.85 mg/mL) compared to purified C-PC in this study. The low values of C-PC concentration in Amarante et al. [1] study can be explained by the adsorption of proteins during filtration and salt addition processes that could cause protein denaturation. This hypothesis needs to be better elucidated by further studies [14] but enlightens the differences in C-PC parameters that could affect antioxidant capacity.

According to Moylan et al. [44], oxidative stress is the cause of multiple cellular dysfunctions promoting diseases such as cancer, atherosclerosis, diabetes, Alzheimer's, chronic heart failure, rheumatoid arthritis, Crohn's disease, and others. All these chronic diseases are often accompanied by muscle wasting. In this respect, the comprehension of C-PC potential to protect the body's cells from damage caused by antioxidant activity as Reactive Oxygen Species and Reactive Nitrogen Species (ROS and RNS) are necessary. The upper levels against ORAC and ABTS can be seen in C-PC and SB, which can categorize these compounds as functional foods which promise health benefits after consuming the product that goes beyond basic nutrition and its content of valuable molecules [45,46].

According to Fratelli et al. [14], the DPPH method is the most used to evaluate the antioxidant activity of C-PC extracts because of estimates besides the reaction rates. However, there is no standardization of the presentation results, which can be demonstrated in percentual of inhibition, $IC_{50}$ (amount of C-PC extract necessary to reduce 50% of the initial radical concentration), and μM TE/g (Trolox equivalent antioxidant capacity assay) which show difficulties in the comparison between different studies. A recent paper designed by Qiao et al. [47] demonstrated that C-PC extract in a concentration of 6.4 mcg/mL presented about 32% of inhibition rate in DPPH. Moreover, this study presented innovative results that showed C-PC encapsulation to form C-PC-Sodium Alginate/Lysozyme complexes with higher scavenging ability, and 40% of inhibition rate against DPPH.

To understand C-PC degradation, it is crucial to understand its structure, which consists of an open-chain tetrapyrrole chromophore (phycobilin) covalently linked to protein molecules [48]. Jaeschke et al. [16] described that C-PC degradation is promoted by oxygen, free radicals, and acids that react with the C-PC tetrapyrrole chromophore, which is responsible for the blue color of the molecule, changes in the chromophore structure which favor color fading, and antioxidant activity losses.

The proximate composition of SB and RB are presented on a wet basis in Table 4. For better literature comparison, the values were converted to % of dry basis: protein content in SB and RB are 52.05% and 61.16%, respectively, that it is in accordance to other *Spirulina* protein content surveys, attesting that *Spirulina* protein can reach from 50% up to 70% on a dry basis when produced under optimum conditions [45], assuring that proteins are the majority *Spirulina* macronutrient as shown in this research.

This is the first paper that included the RB as a by-product of C-PC extraction. Unless RB had higher results than C-PC and SB samples of moisture, ash, and protein, the ORAC ABTS, and DPPH did not accompany this rise. C-PC was extracted from SB, and in the final process, RB was collected, indicating a lower C-PC bioactive compound content and, consequently, lower antioxidant activity. More studies are needed for this assumption.

There are few studies of C-PC effects in vivo, and to the best of our knowledge, this is the first investigation to compare the influence of C-PC, SB, and RB on glucose metabolism and tissue, body, and liver weight. The study that most closely resembles was conducted by Zhao et al. [8] which investigated the anti-obesity effects of whole *Spirulina* (WSP), *Spirulina* protein (SPP equal to C-PC in this study), and *Spirulina* protein hydrolysate (SPPH) in mice fed with high-fat diet (HFD). The potential acting mechanism of SPPH was also explored, which is a little different from this paper's goals, which proposes to understand how each stage of *Spirulina* by-products affects biological and metabolic characteristics of Swiss mice.

Zhao et al. [8] concluded that the anti-obesity effects of SPP (or C-PC) and SPPH are superior to WSP (or SB) under the same dose. Additionally, SPPH demonstrated a good glucose reduction (23.8% of serum glucose) and weight reduction (39.8% of body weight). In Table 7, the lowest body weight results in the final treatment were in HFDC, CD, and HFD and were significantly equal. HFDC and CD differ only from the HFDS and HFDR treatments. That is, the addition of the supplements C-PC, *Spirulina*, and RB did not seem to improve the weight results of the mice, differently from Zhao et al. [8] study.

According to a systematic review and meta-analysis conducted by Bohórquez-Medina et al. [49] including studies with *Spirulina* to understand its impact in adults with obesity, diabetes, or dyslipidemia stated that *Spirulina* supplementation resulted in a de-

crease in triglycerides and total cholesterol levels in these individuals. However, *Spirulina* in fasting blood glucose, glycosylated hemoglobin, and low-density lipoprotein cholesterol levels were not significantly reduced. Additionally, the reduction in anthropometric measures was not significant; these considerations put the utilization of SB, C-PC, and RB as supplements in doubt, so more studies need to be done to establish the period, route, and quantity.

In the present study, as already mentioned, the authors demonstrated biological and metabolic effects in mice, comparing different groups divided in CD, HFD, HFDS, HFDC, and HFDR. However, to understand the C-PC isolated effects, statistical analysis was made to compare CD, HFDS, and HFDC; then, we observed that the delta weight of the HFDS was statistically higher than the others. The comparison between CD, HFDS, and HFDC demonstrated that there was an increase in absolute mass of inguinal adipose tissue (RET, MES, and EPI) and liver in the group supplemented with *Spirulina* biomass when compared to CD and HFDC. Following the presented data, another analysis (data not shown) has been conducted to understand the differences in biological and metabolic effects considering the proximate composition and antioxidant activity.

In 2005, Zhou et al. [50] highlighted the factors that influenced the antioxidant activity of C-PC as *Spirulina*'s major phycobiliprotein (PCB). The authors assumed that C-PC could directly prevent free radical-induced cell damage in humans because of its potential to scavenge radicals. In the present study, the proximate composition did not show remarkable differences between the samples, mainly in protein content. On the other hand, the samples with C-PC isolated and C-PC in *Spirulina* complex matrix (displayed in Table 5 in C-PC and SB samples) had higher antioxidant activity against ORAC, ABTS, and DPPH radicals than RB. It would be interesting to understand the micronutrient content in samples, which can be seen as the fragility of our study. In addition, perceiving C-PC as a bioactive compound that is related to physicochemical and biological properties, the antioxidant activity can be explained by that, which is beyond macro and micronutrient content, and linearity among the composition and biological effect is not always perceived.

## 5. Conclusions

*Spirulina* is a trendy food in many areas of the world because of its nutritional and medicinal properties. Depending on the initial biomass, extracting and recovering food-grade C-PC was possible using a green solvent approach. The RB after the C-PC extraction from *Spirulina* showed a great potential use regarding its composition and antioxidant activity, promoting in this way the sustainable use of the whole biomass. The main protein component of *Spirulina*, C-PC, is a potent free-radical scavenger in the present work. Additionally, the results demonstrated that the supplementation of 500 mg/kg of body weight of *Spirulina* biomass in the HFDS group did not show antiobesogenic potential with an HFD, increasing the delta weight of the animals, and showed glucose intolerance compared to the CD and HFDC groups mainly in 8-week AUC. However, it is essential to conduct further studies to bring other interesting responses regarding C-PC biological in vivo effects. The proximate composition and antioxidant activity can explain these differences in biological and metabolic effects between groups and the stability of bioactive compounds used. Extrapolating these results and considering the amount of 500 mg/kg of body weight to C-PC biological and metabolic effects would be necessary with 30 g of C-PC for a person with 60 kg of body weight. Considering the added value, potential difficulties in the oral supplement acceptance, and the complexity of extraction due to several steps in the process, more studies need to be done to better understand the market needs, population demand, and benefits generated in the process.

**Author Contributions:** Conceptualization, C.F. and A.R.C.B.; methodology, A.R.C.B.; formal analysis, C.F. and A.R.C.B.; investigation, C.F., M.B., A.F.S.-N. and L.M.O.; resources, A.R.C.B., V.V.D.R. and L.M.O.; data curation, C.F. and A.R.C.B.; writing—original draft preparation, C.F., M.B. and A.F.S.-N.; writing—review and editing, A.R.C.B., V.V.D.R. and L.M.O.; visualization, C.F., M.B. and A.R.C.B.; supervision, A.R.C.B.; project administration, A.R.C.B., V.V.D.R. and L.M.O.; funding acquisition, A.R.C.B. All authors have read and agreed to the published version of the manuscript.

**Funding:** This study was financed in part by the "Coordenação de Aperfeiçoamento de Pessoal de Nível Superior—Brasil (CAPES)"—Finance Code 88887.364166/2019-00. This work was also supported by "Fundação de Amparo à Pesquisa do Estado de São Paulo—FAPESP", through the grants process n∘ 2020/07046-0 and 2020/06732-7.

**Institutional Review Board Statement:** The animal study protocol was approved by the Experimental Research Committee of the Federal University of São Paulo (UNIFESP) approved all procedures for the care of the animals used in this study (CEUA-UNIFESP N° 8626090821), approved in 07/01/2021.

**Informed Consent Statement:** Not applicable.

**Data Availability Statement:** The data supporting reported results can be accessed via email for the author (anna.braga@unifesp.br).

**Acknowledgments:** A sincere thanks to "Fazenda Tamanduá®" for the organic powdered *Spirulina* donation after initial biomass selection tests. We also thank the students Stéphanie Fabrícia Francisco da Costa, Leonardo Ribeiro Bernardo, and Monica Masako Nakamoto for their assistance and partnership in laboratory analyses.

**Conflicts of Interest:** The funders had no role in the study's design, in the collection, analyses, or interpretation of data, in the writing of the manuscript, or in the decision to publish the results, declaring no conflict of interest.

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
