# Peer review of "Green Extraction Process of Food Grade C-phycocyanin: Biological Effects and Metabolic Study in Mice"

_processes, doi:10.3390/pr10091793_

Round 1

Reviewer 1 Report

The article "Green extraction process of food grade C-phycocyanin: biological effects and metabolic study in mice" presents interesting results that deserve to be published.

Some small observations can be made:

- In the Tables, the meaning of the letters a, b, c (superscript) is not clearly understood

- The conclusions could be improved with data from the text, as in the Abstract

Author Response

Reviewer 1:

Comment: The article "Green extraction process of food grade C-phycocyanin: biological effects and metabolic study in mice" presents interesting results that deserve to be published.

Some small observations can be made:

- In the Tables, the meaning of the letters a, b, c (superscript) is not clearly understood

- The conclusions could be improved with data from the text, as in the Abstract.

Response: The authors appreciate the opportunity to improve the manuscript. The suggested changes were made.

Reviewer 2 Report

Manuscript ID: processes-1888624

Title: Green extraction process of food grade C-phycocyanin: biological effects and metabolic study in mice

Comments: Having thoroughly reviewed the above manuscript, I have to suggest that this manuscript should be returned with a major revision. Overall, this work is well designed, and shows some interesting results. The authors investigated the different parameters in the green process of organic Spirulina biomass C-PC extraction and further compared the weight and glucose tolerance on Swiss mice fed with a high-fat diet of SB, C-PC, and RB supplementation. However, there are many points unclear in the present manuscript, and the language also need to be improved before publication.

My big doubt is that, as shown in Table 7, no significant difference (p0.05) was observed in both final weight and delta weight among CD, HFD and HFDC, it seems that C-PC rich extract supplementation had no antiobesogenic potential. How could you obtain the conclusion of “the C-PC rich extract improved body weight, decreased white adipose depot weight,” (Line 448)?? On the other hand, no experiments were performed in the present study to determine the white adipose depot. Hence, the conclusion is arbitrary and unconvincing. Also, the same issue in the Abstract section (Lines 28 and 29).

Some following major revisions are suggested for improving the overall quality of the manuscript:

1.   Abstract:

n  Please give the full name of C-PC, ORAC, ABTS, CD, HFD, HFDS, HFDC, HFDR instead of abbreviation, as they all appear for the first time in the Abstract.

n  Line 23, it should be “RM samples”, not “RB samples”.

n  Lines 22 and 23, the authors wrote “The protein content was around 10.7% on a wet basis and, on a dry basis, represents 56.6% in SB and RB samples.” How did you obtain these data? They were not appearing in the following manuscript.

n  Line 24, “smaller”? It should be higher.

n  Please rewrite the Abstract with more detailed data.

2.   Introduction:

n  Please add some similar in vivo studies about C-PC extract in mice.

n  The relationship among the antioxidant activity, biological and metabolic effects should be added.

3.   Materials and Methods 

n  In Figure 2, what’s the meaning of “Coffee and qualitative filter paper……” in “C-PC gathering and filtration”?

n  Line 151, At least three indexes should be performed on the antioxidant activity, only ORAC and ABTS were not enough. Determination of ferrous ion chelating activity, estimation of reducing power, DPPH or FRAP analysis could be added, choose one or more indexes.

n  Line 192, I don’t understand the unit of “500 mg/body weight”, please explain.

4.   Results 

n  Lines 234 and 235, the data are wrong, it should be 47.00%, 47.02%, and 37.14%. Line 267, 88.28% is wrong, also in Line 269, please check.

n  Delete the reference of 14 and 25 in the title of Table 2 and Table 3.

n  Line 271, why don’t you test the proximate composition of C-PC, especially the protein majority? It’s important data.

n  Table 4, the data of Carbohydrate should be reported as mean ± standard deviation.

n  Line 300, the author wrote “HFDS and HFDR differed from CD and HFDC.”

It was incorrect, according to Table 7, HFDR had no significant difference with CD and HFD (p0.05).

n  Give more discussion about Table 8.

n  What’s the unit of y-axis (AUC) in Figure 3?

n  Line 368, “CP-C” should be corrected.

5.   Conclusions

n  As shown in Table 7, no significant difference (p0.05) was observed in both final weight and delta weight among CD, HFD and HFDC, it seems that C-PC rich extract supplementation had no antiobesogenic potential. How could you obtain the conclusion of “the C-PC rich extract improved body weight, decreased white adipose depot weight,” (Line 448)?? On the other hand, no experiments were performed in the present study to determine the white adipose depot. Hence, the conclusion is arbitrary and unconvincing.

n  Why did the proximate composition and antioxidant activity can explain these differences in biological and metabolic effects between groups and the stability of bioactive compounds used? What’s the relationship between antioxidant activity and metabolic effects??

6.  The format of the Materials and Methods section, Results section and the references citation are not according the "Instructions for Authors" of <Processes> journal.

Author Response

Reviewer 2:

Comment: Having thoroughly reviewed the above manuscript, I have to suggest that this manuscript should be returned with a major revision. Overall, this work is well designed, and shows some interesting results. The authors investigated the different parameters in the green process of organic Spirulina biomass C-PC extraction and further compared the weight and glucose tolerance on Swiss mice fed with a high-fat diet of SB, C-PC, and RB supplementation. However, there are many points unclear in the present manuscript, and the language also need to be improved before publication.

My big doubt is that, as shown in Table 7, no significant difference (p>0.05) was observed in both final weight and delta weight among CD, HFD and HFDC, it seems that C-PC rich extract supplementation had no antiobesogenic potential. How could you obtain the conclusion of “the C-PC rich extract improved body weight, decreased white adipose depot weight,” (Line 448)? On the other hand, no experiments were performed in the present study to determine the white adipose depot. Hence, the conclusion is arbitrary and unconvincing. Also, the same issue in the Abstract section (Lines 28 and 29).

Response: The authors appreciate the opportunity to improve the manuscript. The suggested changes were made. In the present study the authors demonstrated biological and metabolic effects in mice comparing different components divided in CD, HFD, HFDS, HFDC and HFDR groups. However, to understand the C-PC isolated effects, statistical analysis was made to compare CD, HFDS and HFDC then, we observed that the delta weight of the HFDS was statistically higher than the others. The comparison between CD, HFDS and HFDC demonstrated that there was an increase in absolute mass of inguinal adipose tissue (RET, MES and EPI) and liver in the group supplemented with Spirulina biomass when compared to CD and HFDC.

Comment: Abstract: Please give the full name of C-PC, ORAC, ABTS, CD, HFD, HFDS, HFDC, HFDR instead of abbreviation, as they all appear for the first time in the Abstract. Line 23, it should be “RM samples”, not “RB samples”. Lines 22 and 23, the authors wrote “The protein content was around 10.7% on a wet basis and, on a dry basis, represents 56.6% in SB and RB samples.” How did you obtain these data? They were not appearing in the following manuscript. Line 24, “smaller”? It should be higher. Please rewrite the Abstract with more detailed data.

Response: The explanation of the attainment on wet and dry basis were included on Methods in lines 213 to 216. The term higher was included to correction. The full name of C-PC, ORAC, ABTS, CD, HFD, HFDS, HFDC, HFDR were added to the abstract, the abbreviations were used to respect the 200 words abstract limitation. The correct abbreviation was provided: Remaining biomass (RB), and all changes were made in the manuscript regarding this abbreviation.

Comment: Introduction: Please add some similar in vivo studies about C-PC extract in mice. The relationship among the antioxidant activity, biological and metabolic effects should be added.

Response: The theoretical foundation about C-PC extract in mice was included in Introduction, as well as the relationship among biological and metabolic effects of this bioactive component.

Comment: Materials and Methods: In Figure 2, what’s the meaning of “Coffee and qualitative filter paper……” in “C-PC gathering and filtration”? Line 151, At least three indexes should be performed on the antioxidant activity, only ORAC and ABTS were not enough. Determination of ferrous ion chelating activity, estimation of reducing power, DPPH or FRAP analysis could be added, choose one or more indexes. Line 192, I don’t understand the unit of “500 mg/body weight”, please explain.

Response: Figure 2 was corrected as Material and Methods description. The nomenclature “500 mg/body weight” has been changed by “500 mg/kg of body weight” in line 267. This means that the supplementation amount standardization was dependent on Swiss mice weight, that is, 500 mg of Spirulina, RB or C-PC per each kg of animal weight. The DPPH analysis was included as requested.

Comment: Results: Lines 234 and 235, the data are wrong, it should be 47.00%, 47.02%, and 37.14%. Line 267, 88.28% is wrong, also in Line 269, please check. Delete the reference of 14 and 25 in the title of Table 2 and Table 3. Line 271, why don’t you test the proximate composition of C-PC, especially the protein majority? It’s important data Table 4, the data of Carbohydrate should be reported as mean ± standard deviation. Line 300, the author wrote “HFDS and HFDR differed from CD and HFDC.” It was incorrect, according to Table 7, HFDR had no significant difference with CD and HFD (p>0.05). Give more discussion about Table 8. What’s the unit of y-axis (AUC) in Figure 3? Line 368, “CP-C” should be corrected.

Response: The Lines 234 and 235 were wrong, the calculation of percentage change, % change = (higher value – lower value / lower value) * 100, is the best way to demonstrate the increment of concentration, yield, and purity between SBV and SFT, because of that, the values were arranged. The same modification of percentage change were made in lines 350, 478and . The standard deviation of Carbohydrates was included. The protein majority of C-PC was included in the study as suggested, the determination was performed using Bradford method. The authors rewrite the line 300 to better comprehension about the final weight after treatments, demonstrating that HFDS, HFDR, and HFD groups showed the highest weight values and HFDS and HFDR differed from CD and HFDC. The discussion about Table 8 were incorporated. The unit of y-axis (AUC) in Figure 3 refers to the area under the curve (AUC). Line 368, “CP-C” was corrected as requested.

Comment: Conclusions: As shown in Table 7, no significant difference (p>0.05) was observed in both final weight and delta weight among CD, HFD and HFDC, it seems that C-PC rich extract supplementation had no antiobesogenic potential. How could you obtain the conclusion of “the C-PC rich extract improved body weight, decreased white adipose depot weight,” (Line 448)?? On the other hand, no experiments were performed in the present study to determine the white adipose depot. Hence, the conclusion is arbitrary and unconvincing. Why did the proximate composition and antioxidant activity can explain these differences in biological and metabolic effects between groups and the stability of bioactive compounds used? What’s the relationship between antioxidant activity and metabolic effects??

Response: In the present study the authors demonstrated biological and metabolic effects in mice comparing different groups divided in CD, HFD, HFDS, HFDC and HFDR groups. However, to understand the C-PC isolated effects, statistical analysis was made to compare CD, HFDS and HFDC then, we observed that the delta weight of the HFDS is statistically higher than the others. The comparison between CD, HFDS and HFDC demonstrated that there was an increase in absolute mass of inguinal adipose tissue (RET, MES and EPI) and liver in the group supplemented with Spirulina biomass when compared to CD and HFDC. Further analysis, which are not included in the current paper scope (data not shown), have been conducted to understand the differences in biological and metabolic effects considering the proximate composition and antioxidant activity.

In 2005, Zhou and collaborators highlighted the factors that influenced antioxidant activity of C-PC, as the major phycobiliprotein (PCB) from Spirulina, the authors assumed that C-PC could directly prevent free radical-induced cell damage in humans because its potential of scavenge radicals. In the present study, the proximate composition did not show great differences between the samples mainly in protein content. On the other hand, the samples with C-PC isolated and C-PC in Spirulina complex matrix (showed in Table 5 in C-PC and SB samples) had higher antioxidant activity against ORAC, ABTS and DPPH radicals than Remaining biomass. It would be interesting to understand the micronutrients content in samples, that can be seen as a fragility of our study. Besides that, thinking about the C-PC as a bioactive compound, that is related to physicochemical and biological properties, the antioxidant activity can be explain by that, that is beyond macro and micronutrients content.

Reference:

ZHOU, Z. P., LIU, L. N., CHEN, X. L., WANG, J. X., Chen, M. I. N., ZHANG, Y. Z., & ZHOU, B. C. (2005). Factors that effect antioxidant activity of C‐phycocyanins from Spirulina platensis. Journal of Food Biochemistry, 29(3), 313-322.

Comment: The format of the Materials and Methods section, Results section and the references citation are not according the "Instructions for Authors" of <Processes> journal.

Response: The paper has been placed in the standards according to the available Processes template. However, to be in agreement with the reviewer's suggestions some changes and additions to the formatting were made in Material and Methods, Results and References citation sections.

Round 2

Reviewer 2 Report

Authors have revised the manuscript step by step according to my comments. Manuscript is more acceptable now.